# ADHD Follow-Up in Adulthood among Subjects Treated for the Disorder in a Child and Adolescent Mental Health Service from 1995 to 2015

**DOI:** 10.3390/medicina59020338

**Published:** 2023-02-10

**Authors:** Rosaria Di Lorenzo, Jessica Balducci, Anna Cutino, Emanuela Latella, Giulia Venturi, Sergio Rovesti, Tommaso Filippini, Paola Ferri

**Affiliations:** 1Psychiatrist of Department Mental Health and Drug Abuse, AUSL-Modena, 41126 Modena, Italy; 2Psychiatrist of Department Mental Health and Drug Abuse, AUSL-Bologna, 40131 Bologna, Italy; 3Psychiatrist of Department Mental Health and Drug Abuse, ASP-Trapani, 91100 Trapani, Italy; 4School of Specialization in Psychiatry, Department of Biomedical, Metabolic and Neural Sciences, University of Modena and Reggio Emilia, 41125 Modena, Italy; 5Section of Public Health, Department of Biomedical, Metabolic and Neural Sciences, University of Modena and Reggio Emilia, 41125 Modena, Italy; 6School of Public Health, University of California Berkeley, Berkeley, CA 94704, USA

**Keywords:** adult ADHD, follow-up ADHD in adulthood, ADHD comorbidities

## Abstract

*Background and Objectives:* ADHD is a neurodevelopmental disorder characterized by inattention and hyperactivity/impulsivity and can persist in adulthood. The aim of this study is to deepen knowledge about adult ADHD follow-up. *Materials and Methods:* This observational study consists of one retrospective part aimed at collecting records of children and adolescents treated for ADHD in the Children and Adolescent Mental Health Service (CAMHS) from 1995 to 2015 and, successively, at identifying their adult follow-up in Adult Mental Health Service (AMHS); the second part consists of ADHD scale administration, Diagnostic Interview for ADHD in Adults (DIVA 2-0) and Adult Self Rating Scale (ASRSv1.1), for the subjects currently being treated at AMHS who agreed to participate in the study. *Results:* We observed that among the 55 patients treated at CAMHS between 1995 and 2015 for ADHD and subsequently at the AMHS, none presented a diagnosis of ADHD; instead, they were treated for Intellectual Dysfunction (33%), Borderline Personality Disorder (15%) and Anxiety Disorders (9%), and two individuals were also diagnosed with comorbid substance/alcohol abuse (4%). Of the 55 patients, only 25 (45%) were treated at AMHS during the study period. Though we asked for their informed consent to administer the questionnaires, we were able to test only seven patients. The ASRS-V1.1 score showed that 43% of patients reported symptoms of ADHD persistence in adulthood. For DIVA 2.0, 57% of individuals reported scores indicating the persistence of the ADHD inattention component, and 43% the persistence of both ADHD dimensions. *Conclusions*: ADHD cannot be considered a disorder confined to childhood/adolescence but instead is a chronic and complex condition that can persist into adulthood. The very small size of our final sample may account for both the high ADHD dropout rate over the long follow-up period and the difficult transition from child to adult health care in ADHD treatment. Our investigation suggests the need for specific training in the diagnosis and treatment of adult ADHD and the implementation of transition protocols between minor and adult services to improve long-term treatments.

## 1. Introduction

Attention-deficit/hyperactive disorder (ADHD) is one of the most common neurodevelopmental disorders. It normally presents two symptom dimensions: inattention and impulsivity/hyperactivity [1,2]. The disorder is more frequent in males, with a ratio of 2:1, which tends to balance in adulthood. Females with ADHD mainly manifest inattentive symptoms and tardive clinical manifestation, though this disorder can be misdiagnosed in females [3,4]. ADHD occurs with a prevalence ranging between 5% and 7% [5,6] in childhood and persists into adulthood in 15% to 65% of patients [3]. In accordance with many authors, adult persisting ADHD is a chronic and debilitating disease, with poor quality of life and socio-economic status, low probability of a stable marriage and a long-term job, and higher probability of hospitalization, forensic issues, marginality, and mortality [4,7,8].

Although the first scientific description of ADHD was given by a pediatrician, George Frederic Still, in 1902 [9], only in 1987 did the DSM-III-R introduce the term ADHD, indicating the two core symptoms of hyperactivity and inattention, and define a residual type with a clinical persistence in adulthood [10]. Following this, the DSM-5 introduced further criteria for clinical presentation of ADHD in adulthood as well as the reduction from six to five symptoms required for diagnosis in people aged from 17 years to adulthood [11,12,13]. Increasing evidence shows that about half of patients with ADHD have persisting ADHD symptoms into adulthood [14,15], although clinical manifestations of this disorder change with increasing age.

The pathogenesis of ADHD is multifactorial and conditioned by genetic vulnerability [16,17,18], environmental factors [19], prenatal conditions, prematurity [20,21] and neurobiological alterations, mainly involving prefrontal cortex (PFC) impairment [22]. Shaw and colleagues [23] highlighted that a delayed maturation of PFC is present in adolescents with ADHD and that it is linked to symptom persistence into adulthood, whereas the thinness of bilateral prefrontal areas seems to correlate with the severity of conduct disorder [24].

### 1.1. ADHD in Adulthood

In adulthood, prevailing symptoms are inattention, restlessness and disorganization in daily activity but not hyperactivity, while impulsiveness can persist [10,15,25,26]. Therefore, specific diagnostic criteria for ADHD in adulthood could improve diagnostic detection [27]. Diagnosis of ADHD is generally based on clinical observation and anamnesis from multiple sources (e.g., parents, teachers) and also uses semi-structured interviews and standardized questionnaires. In the case of adult patients, the history of the evolution of symptoms and the level of functioning during adolescence is of utmost importance to acknowledge undetected ADHD manifestation [28]. Moreover, collecting collateral information from parents or caregivers [29], considering that the memory of childhood symptoms can vary considerably among adult patients, can increase the accuracy of diagnosis [27]. In the case of clinical suspicion, the administration of screening scales such as the Adult ADHD Self-Report Scale (ASRS-V1.1) Symptom Checklist can be useful [30].

For completing the diagnostic investigation, the Diagnostic Interview for Adult ADHD, which is available in the 5th version (DIVA-5) as semi-structured questionnaire, is a valid and reliable tool [31,32]. To date, research has increasingly shown evidence that ADHD is a lifelong chronic disease, and authors have also suggested a late-onset type [33,34]. Nowadays, uniform epidemiological data are still lacking [35].

The clinical manifestation of associated disorders as well as ADHD sub-threshold symptoms can lead to misdiagnosis, even if more recent research shows a trend of increasing numbers of diagnoses in adulthood, suggesting changes in clinical practice [36]. Therefore, healthcare professionals could explore ADHD symptoms in patients with mainly behavioral and cognitive manifestations, considering the negative prognostic consequences of a delayed diagnosis [37]. Indeed, if consensus in the literature states that the most severe and impairing clinical illness must be addressed first, it is true that early treatment of ADHD reduces the severity of psychiatric comorbidities and leads to better clinical course [33,35,38].

### 1.2. ADHD Comorbidities

ADHD is often associated with one or more comorbid psychiatric disorders, leading to worse prognosis and more complex treatments [39]. More common comorbid disorders, often characterized by early occurrence and clinical severity, are represented by anxiety and affective disorders, personality disorders, substance use disorder (SUD) and eating disorders [25,40,41,42,43]. Mood and anxiety disorders, depression and bipolar I disorder affect a significant proportion of patients with ADHD [33]. SUD affects ADHD patients at twice the frequency of the general population and is associated with poly-substance use, frequent hospitalizations, suicidal behavior, and low compliance with treatment [44]. Antisocial personality disorder, which is in continuity with oppositional defiant disorder, conduct disorder and depression during childhood/adolescence, is often associated with ADHD and SUD, therefore representing a negative prognostic factor, due to impaired functioning during adult life [45]. Notably, ICD-10 [46] distinguished a “Hyperkinetic conduct disorder” from ADHD due to the different prognosis [47]. ADHD patients with personality disorders show poorer responses to pharmacological treatment and poorer compliance with therapy than patients without personality disorders [48].

### 1.3. ADHD Therapeutic Issues

Symptoms of ADHD are linked to abnormalities in dopamine and norepinephrine signal pathways, which constitute the main therapeutic target of psychopharmacological drugs [49]. In fact, the most widely used drugs are psychostimulants, used mainly for childhood treatment of severe ADHD [50], even if the prescription of these therapies is widely extended to late adolescence and adulthood [51,52]. A systematic review highlighted that this treatment could change brain structures and signaling, making them similar to non-ADHD patients [53]. Psychostimulant treatment improves neuropsychological skills, such as executive functions, memory, impulse control, attention and cognitive performance [54,55,56], which have a significant impact on quality of life and functioning [34,57,58]. Nonetheless, treatment of ADHD must include non-pharmacological treatments, as suggested by some authors [59] and guidelines [60,61,62]. Multimodal approaches, including psychotherapeutic and pharmacotherapy treatments, have demonstrated good outcomes in adult patients with persisting symptoms [63,64,65]. NICE guidelines also highlight the importance of building a comprehensive transitional protocol between mental health services for children and adults [60] due to the risk that premature interruption of treatment and/or future misdiagnosis and inappropriate treatments favor chronicity [25].

### 1.4. Aims

The aim of this study is to deepen knowledge about ADHD follow-up in adulthood in order to improve its clinical management from childhood to adulthood. After having conducted a scoping review on the persistence of ADHD in adulthood and its complications [7], we investigated the clinical outcomes of subjects treated by the Child and Adolescent Mental Health Service (CAMHS) in a Northern Italian town from 1995 to 2015, who continued to be treated during adulthood by the Adult Mental Health Service (AMHS).

## 2. Materials and Methods

### 2.1. Study Design

This observational naturalistic study presents both a retrospective and prospective design. It consists of two parts: the first retrospective one was aimed at collecting records of children and adolescents treated for ADHD in the CAMHS in Modena from 1995 to 2015 and, successively, at identifying their adult follow-up in AMHS, collecting demographic and clinical information about individuals treated from services databases; the second part consists of ADHD scale administration to the subjects currently being treated at AMHS who agreed to participate in the study, giving us their informed consent.

This study received the approval of the Ethics Committee of the Area Vasta Emilia Nord (Prot. No. AOU0007877/19 of 19/03/2019 and subsequent amendment Prot. No. AOU0024912/20 of 09/15/2020) and was authorized by the Modena Local Health Authority (Decision No. 658 of 26/03/2019).

The primary objective of the study was to determine how many patients were still being treated in AMHS among those treated for ADHD as minors from 1995 to 2015 in CAMHS, evaluating the demographic and clinical factors that could have influenced the evolution of ADHD in patients treated in CAMHS from 1995 to 2015: single parenting, adoption, age of treatment start at CAMHS, schooling, intellectual disability, the presence of comorbidities and types of treatments.

The secondary objective was the evaluation of ADHD persistence in adulthood by means of scale administration, and the presence or absence of psychiatric and substance use comorbidities.

### 2.2. Exclusion and Inclusion Criteria

All patients with a diagnosis of ADHD (ICD-10: F.90.0 and F.90.1) [46], who were treated in CAMHS from 01/01/1995 to 31/12/2015 and were born before 2002, were included in the retrospective study, and all patients from that group currently being treated in AMHS, who agreed to participate in the study, giving their written consent, were included in the prospective part of the study. Patients who could not give their valid consent or who were still minors at the time of the data extrapolation were excluded.

### 2.3. Data Collection

The variables were selected to describe the demographic and clinical characteristics of our sample both in childhood and in adulthood among those available in the information system of outpatient services. Data collected from the CAMHS information system were the following: single parenting, adoption, age at the time of treatment start in CAMHS, referral to CAMHS, possible coexistence of intellectual disability or other comorbidities and types of treatments (pharmacological and not).

We retrospectively collected the following variables from the AMHS information system: psychiatric diagnosis and substance use, period of treatment, schooling, employment, housing, hospitalizations, treatments, social service support and legal problems. More information was sought from the referral psychiatrist of the adult individuals treated at AMHS.

The prospective part was aimed at the evaluation of ADHD persistence in adulthood. It consisted of the administration of two questionnaires validated in Italian, the Adult Self-Report Scale (ASRS-v1.1) and Diagnostic Interview for ADHD in Adults (DIVA-2.0), after having received patients’ written consent.

### 2.4. Rating scales

#### 2.4.1. ASRS-V1.1

The Adult Self Rating Scale (ASRS-V1.1) [30] investigates ADHD symptoms in accordance with the 18 criteria of DSM-IV-TR. Six of the eighteen questions are included in Part A of the List of Symptoms and are found to be the most predictive for ADHD. It takes about 5 min to complete the list, and this can provide information that is essential for integrating the diagnostic process. Part B of ASRS-V1.1 contains the remaining twelve questions, which help with ADHD screening in adult patients. Knowledge gained through the ASRS-V1.1 screening may suggest the need for a more in-depth clinical interview.

#### 2.4.2. DIVA 2.0

The Diagnostic Interview for ADHD in Adults (DIVA) is based on the criteria of the DSM-IV and is the first Dutch structured interview for ADHD in adults, developed by Kooij and Francken [66]. The interview is divided into three parts, each of which is aimed at both children and adults:the criteria for Attention Deficit (A1);the criteria for hyperactivity / impulsivity (A2);the age of onset and the dysfunction caused by the symptoms of ADHD.

Each of the 18 criteria described by DSM-IV is investigated one at a time, first assessing the presence in adulthood and then the possible presence in childhood.

In order to make it easier to verify the presence or absence of the 18 symptomatic criteria of ADHD, both in childhood and adulthood, the interview provides a list of concrete and realistic examples, referring to both current (adult) and child behavior. These examples are taken from the most frequent descriptions given by adult patients in clinical practice and concern the difficulties typically associated with symptoms in five areas of daily life: work and education, love and family relationships, social relationships, leisure and hobbies, self-esteem and self-image. If possible, DIVA 2.0 should be administered to adults in the presence of the partner and/or a family member, to obtain more information. DIVA 2.0 administration takes about one hour/one hour and a half. DIVA 2.0 only investigates the key symptoms of ADHD required to diagnose ADHD according to the DSM-IV and does not investigate any symptoms, syndromes or psychiatric disorders present in comorbidities.

### 2.5. Statistical Analysis

Data were statistically analyzed and included mean ± Standard Deviation (SD) and percentage for descriptive analysis. For the comparison between means, Student’s *t*-test was applied for unpaired data and the chi-squared test was used to compare percentages.

## 3. Results

We retrospectively collected 303 patients diagnosed with ADHD, in accordance with ICD-10, from the CAMHS database (Figure 1). Of these, only 55 individuals were referred to AMHS in adulthood, whereas others were discharged or self-discharged from CAMHS, referred to other services or transferred to other towns/countries, as shown in Table 1.

We excluded from the study the total number of individuals who were not treated at local CAMHS (n = 237) and those who were exclusively treated at Substance Use Services (SUS) (n = 11) (Figure 1).

From the sample of children/adolescents treated at CAMHS (n = 303), 55 were treated in adulthood at AMHS (18%) and 11 (4%) at Substance Misuse Service; 7 (2%) individuals among them were taken in care by both services (Figure 1).

### 3.1. Demographic Characteristics of the Sample

Most of our sample was composed of males (71%) and Italians (84%). Regarding the demographic characteristics, 44% of the sample had obtained a secondary school diploma, 35% a lower secondary school diploma and only 6% a university degree. Currently, 38% of individuals were employed in a protected job, 16% were engaged in studies, 15% were employed in a regular job, 13% were unemployed and a further 13% received a disability pension. Most individuals lived with their family of origin, composed of both biological parents in 56% of cases, a single parent in 9% of cases, adoptive parents in 6% of cases and foster parents in 2% of cases; 13% lived in a protected apartment or in a community, 4% with their acquired family and 2% alone.

### 3.2. Clinical Characteristics of the Sample

Analyzing the childhood clinical variables of subjects treated at CAMHS for ADHD and subsequently at AMHS in adulthood (n = 55), we highlight the following results:60% of the individuals were taken into care before the age of 5;in 42% of cases, the referral to CAMHS was made by school;most of the minors were treated at CAMHS for a long period of time, ranging between 10 and 15 years in 49% of cases;53% of individuals were treated with only rehabilitative intervention and 47% with multimodal (pharmacological and rehabilitative) treatment;29% of individuals were also in care at Social Service for Minors;89% were discharged from CAMHS at an age ranging between 18 and 21 years;49% of individuals were taken into care by Adult Mental Health Service at the age of majority;58% of the patients treated in CAMHS for ADHD had a psychiatric comorbidity, whereas 27% a neurological comorbidity and 9% a congenital/hereditary type.

The psychiatric diagnosis most frequently associated with ADHD in childhood was found to be intellectual disability (44% of cases), which was mild in 54% of individuals, moderate in 38% and severe in 8%. The diagnosis of ADHD was associated with specific developmental disorders of school skills (50% of cases), conduct and emotional disorder (29%), oppositional-defiant disorder (18%), anxiety-depressive disorders (13%), Asperger Syndrome (7%), unspecified manic episodes (4%) and obsessive-compulsive disorder (4%). In a percentage of 2% for each, we found a concomitant diagnosis of Schizotypal Personality Disorder, Post-Traumatic Stress Disorder, Cyclothymia, Gilles de la Tourette Syndrome and Autism.

As regards the analysis of the adulthood clinical variables of our sample treated at AMHS (n = 55), we found the following:45% of individuals were still being treated at AMHS (n = 25);the majority (n = 36; 65%) of adult individuals appeared to have had their first access to AMHS at an age ranging between 18 and 20 years, 24% of patients between 21 and 24 years, and 6% under 18 years;in 49% of cases, the referral to AMHS was CAMHS, in 27% a general practitioner, in 9% spontaneous access, 7% though Social Services, and 6% other sources (judicial referral or at hospital discharge);15% of patients were hospitalized, with a single case (2%) being compulsory hospitalization;15% of patients were sent to a community treatment;7 individuals (13%) had also been treated by SUS and successively discharged;70% of individuals were also in care at Social Services (n = 38);62% of the sample was treated with drug therapy and socio-rehabilitation interventions (n = 34);15% of the sample was treated with a psychotherapy (n = 8);7% of individuals were found to have had legal problems (n = 4);3 individuals were legally supported by a so-called legal administrator.

As regards the diagnoses of adult patients in our sample, we found the following:no patient was diagnosed with ADHD;the diagnosis of intellectual disability represented the most frequent diagnosis (24% of the sample);other diagnoses were represented by borderline personality disorders (15%), bipolar disorders (12%), anxiety disorders (9%), schizophrenia spectrum disorders (8%), maladjustment disorders (7%), autism (4%), alcohol/other substance addiction (4%), obsessive-compulsive disorder (2%) and unspecified non-psychotic mental disorders following organic brain damage (2%);7 patients (13%) were evaluated at AMHS, but no psychiatric diagnosis was found during consultation;only 12 individuals (22%) presented non-psychiatric medical comorbidities, in particular epilepsy (11%) and obesity (4%).

Of the 55 patients treated in adulthood at AMHS, 25 were currently in care during the study period in 2021 (Figure 1).

### 3.3. Evaluation of ADHD Persistence in Adulthood

We contacted all patients still in care at AMHS (n = 25) to evaluate the persistence of ADHD by means the administration of DIVA 2.0 and the ASRS-V1.1. As shown in Figure 1, it was not possible to complete the questionnaires for 18 patients because

they were not traceable (n = 5);they did not provide informed consent (n = 3);they were not evaluable due to moderate/severe intellectual dysfunction (n = 10).

The remaining seven patients accepted the administration of the scales (Table 2).

The ASRS-V1.1 self-assessment scale scores showed that three of seven patients (43%) reported symptoms of persistent ADHD in adulthood.

From the administration of DIVA 2.0, we report the following:4 subjects (57%) did not obtain a positive score for adult ADHD;3 subjects (43%) reported symptoms of both ADHD components (inattention and impulsivity/hyperactivity);1 subject (14%) reported symptoms of inattention but not of impulsivity/hyperactivity.

All but one of the patients who had a DIVA 2.0 positive score in at least one of the two ADHD dimensions had a similarly positive score in ASRS-v1.1, indicating that most had a subjective awareness of this disorder.

When comparing the socio-demographic and clinical variables of adult individuals with positive and negative DIVA 2.0 scores, we did not appreciate statistically significant differences between the two groups, except for two variables. More individuals with positive scores at one or two dimensions of DIVA 2.0 in adulthood had required social support in childhood (Pearson chi^2^ = 3.73; *p* = 0.05) compared with those with a negative score, whereas individuals with a negative score for both dimensions of DIVA 2.0 (n = 3) were older than those with positive DIVA 2.0 scores (normal age distribution in Shapiro–Wilk test, V = 0.14, *p* = 0.99; t = 2.85; *p* = 0.03; *t*-test) (Table 2). Moreover, more individuals with positive scores in one or two dimensions of DIVA 2.0 in adulthood were unemployed or had civil invalidity, had needed to be treated in therapeutic communities and suffered from psychiatric disorders as a main or comorbid diagnosis (Table 2).

## 4. Discussion

This study aimed to evaluate the ADHD follow-up from childhood to adulthood in a real-world setting in order to highlight the percentage of adult ADHD persistence and association with psychiatric and substance use comorbidities as well as social problems. In the literature, studies on ADHD in childhood have allowed us to develop shared and comprehensive guidelines, whereas, up to now, the available epidemiological studies on adult ADHD have reported discordant results [8,67].

The retrospective part of this study included 55 individuals followed in childhood at the CAMHS for ADHD and subsequently treated at AMHS in the same Northern Italian town. Of these, 71% were males, data in line with the literature, which highlights a prevalence in males with a 2:1 ratio [12]. The 15-year follow-up period was long enough to highlight the age of onset of this disorder in childhood (mainly less than 5 years) and the onset age of psychiatric comorbidities in adulthood (mainly between 18 and 20 years), data overlapped that reported in the literature [12,68].

Our analysis highlights that only 18% of minors treated for ADHD were subsequently treated by the Adult Mental Health Service on direct indication from CAMHS, 78.9% were discharged or had stopped treatment and 14.3% were self-discharged from CAMHS. This figure is in line with the high dropout rate of subjects with ADHD in the post-adolescent period and testifies to the difficulties relating to the transition from child to adult care [69]. In accordance with some authors [9], we can explain these data by the scarcity of services and professionals specializing in ADHD diagnosis and treatment in adults as well as the difficult transition from child to adult health services.

Our study did not retrospectively report any diagnosis of ADHD in our adult sample, although test scores highlighted that more than half of our patients presented ADHD symptoms in adulthood, further suggesting the difficulties in diagnosing adult ADHD. In this regard, many authors [68,69] put in evidence that adult ADHD diagnosis can be underreported due to many factors: there are no biological markers, so that the diagnosis can be influenced by cultural factors, as suggested by the differences in the prevalence of ADHD worldwide; there are no fully standardized diagnostic criteria; the high rates of comorbidity in adulthood complicate its recognition; physicians with specific training in ADHD in adulthood are not widespread.

The prospective part of our study, which analyzed the prevalence of ADHD diagnosis in adulthood through the administration of ASRS-v1.1 and DIVA.2 scales, found the persistence of inattention symptoms in 57% of individuals, whereas both components of this disorder (inattention and impulsivity-hyperactivity) were maintained in 43% of individuals. These results are in line with what is reported in the literature: about 57% of adults with a clinical history of ADHD in childhood meet the diagnostic criteria of the disorder in adulthood, whereas 66% have more than one ADHD symptom causing discomfort and impaired functioning [36]; in adulthood, the prevalent clinical features are inattention and organizational difficulties, while hyperactivity and impulsiveness become less visible with increasing age [70].

Furthermore, we observed that all our patients, except one, obtained comparable scores for the ASRS-v1.1 to those reported for DIVA 2-0, indicating that most had subjective awareness of this disorder.

The administration of DIVA 2.0 found that two of seven individuals did not obtain a suggestive score for childhood ADHD in either dimension, despite having been followed up for this disorder at CAMHS. This discrepancy could be due to a difficulty in remembering some of the information requested in the diagnostic interview. In fact, to overcome this problem, the presence of a family member is required, where possible, in order to integrate the lack of retrospective information.

It should also be noted that no patient treated due to ADHD at a minor age had maintained this diagnosis in adult mental health services, even in the cases that presented scores suggesting symptoms of ADHD inattention and/or impulsivity on the questionnaires administered. This data could indicate the difficulty in diagnosing a clinical course not yet well defined in adulthood by the current classification systems or also indicate different types of this disorder as hypothesized by some authors [71].

The retrospective study revealed that no individual in the sample continued treatment with methylphenidate or atomoxetine in adulthood. A multicenter study conducted in a sample of adolescents with ADHD documented a 62% reduction in the use of drugs in the age group between 17 and 21 years [72]. The National Institute for Clinical Excellence guidelines [60] emphasize the importance of early diagnosis and continuity of care for people with ADHD in order to prevent its long-term consequences. In the same guidelines, a specific section dedicated to the transition from childhood to adulthood of individuals with ADHD shows the steps necessary to avoid even a brief interruption of treatment, based on re-evaluation in youth for continued treatment into adulthood, new overall assessment of personal, occupational and social functioning and any co-morbidities in adult age, a non-strict age threshold for transition, and annual meeting with the care team, patient and family to review the treatment.

In our sample, the psychiatric comorbidity more frequently associated with ADHD was Intellectual Disability, both in childhood (44%) and in adulthood (33%). The literature shows heterogeneous data in this regard, since the disorder itself involves a deficit of executive functions. Patients with ADHD show, in fact, deficits in numerous cognitive domains, particularly in the attentional and decision-making processes. Similar cognitive difficulties have also been observed in adults [73,74]. In patients with high IQ, poor performance on tests may indicate impaired executive functions, even in the absence of clear deficits. Studies on ADHD are usually conducted on patients with a medium or high IQ level, and the variable of the intellectual level is not always controlled in the interpretation of the results [9]. The high frequency of individuals with intellectual disability in our final sample can suggest that the most fragile and vulnerable subjects, who need the regular support of services for daily activities, maintain the continuity of care more frequently than other individuals with ADHD.

Other disorders frequently associated with ADHD in childhood were Conduct Disorders (29% of cases), specific developmental disorders of school skills (25%) and Oppositional-Provocative Disorder (18%). Similar data emerged from the scoping review [7], in which 15 studies found that the main comorbid disorder with ADHD in childhood is Oppositional Defiant Disorder (9–81%), followed by Conduct Disorder (26–38.3%). These data are in line with the epidemiological evidence found by other authors [75], which suggests a frequent association with disorders characterized by externalizing symptoms and various maladjustment problems throughout life [76]: disciplinary problems and school failures [77], antisocial behaviors, substance use [78,79,80,81], accidents, unemployment and instability in relationships [68].

Criminal activities and judicial problems were found in 7% of our sample. The scoping review highlighted an association between ADHD arising in childhood and crime in adulthood in 32% of cases [7], a figure that is closest to what emerged in the test results, in which 25% of individuals with a suggestive ADHD score at the tests presented problems with the justice system.

Our study observed that most individuals in our sample were currently engaged in a protected job supported by AMHS (38%), while few individuals (15%) had independently achieved work stability. Among the individuals with rating scale scores positive for adult ADHD, the data are even more evident: 50% worked in a protected environment, and none were employed in regular work. These data, which are in accordance with other study results [82], further highlight how ADHD, especially if associated with intellectual disability and/or altered behavior, can heavily affect the evolution of relational, work, and social skills. In our study, the social difficulties developed by individuals diagnosed with ADHD in childhood can also be deduced by the high percentage (70%) who needed assistance from Social Service in adulthood.

The very small size of our final sample can represent the difficult transition between child and adult care for complex disorders previously considered uniquely confined to minors, such as ADHD, which survive into adulthood. In particular, for a person with ADHD, it can be difficult to maintain continuity of care due to hyperactivity/impulsivity and inattention that can predispose one to both therapy dropout and social-relational maladjustment. The adolescents with ADHD “lost in transition” can undergo negative outcomes in adulthood, as highlighted by many authors [83,84].

Transition care is a “multidimensional and multidisciplinary process” [83], which should address many different needs, including social and educational issues [85]. Health services should provide tailored programs to adolescents with ADHD and their family/caregiver in order to address informational, occupational and social needs and not only pharmacological and psychological treatments [86].

### Limitations and Advantages of the Study

This research has several limitations, the main one being the very small size of the final sample due to many factors: the long follow-up period, the high dropout rate in ADHD treatment as reported by literature [87], the difficult transition from child to adult health services and the lack of specialized centers for the diagnosis and treatment of ADHD in adults [9,61]. Moreover, only a few patients (n = 7) were assessable through the scale administration and/or were available to participate in this study, providing us their informed consent. This progressive reduction of the sample to a very limited number of individuals may represent the limiting problem of follow-up studies in a real health context. Furthermore, the data collection of our sample in childhood treatment was limited by the lack of clinical information, such as the type of drug and/or rehabilitative treatment undertaken in childhood or the scales used for ADHD diagnosis, due to incomplete clinical profiles available in the databases consulted.

However, the follow-up design of this study and its implementation in a real-world health setting allowed us to report a realistic course of ADHD from childhood to adulthood in a group of individuals treated in both CAMHS and AMHS and to highlight the difficulties in diagnosing and treating ADHD in the transition between minor and adult services. The careful diagnostic process for ADHD in adults, including both a screening instrument (ASRS v1.1) and a dedicated diagnostic interview (DIVA-2), can be mentioned as a relevant point of strength of the present work.

## 5. Conclusions

This study highlighted that 18% of individuals treated for ADHD in childhood successively needed to be treated by Adult Mental Health Services, in particular due to Intellectual Dysfunction, Borderline Personality Disorder and Anxiety Disorders. Among individuals currently treated in AMHS who agreed to be screened for adult ADHD, we highlight a persistence of at least one of two ADHD dimensions in more than half of cases, although none of these individuals were ADHD diagnosed at AMHS, confirming the difficulty in diagnosing this disorder in adulthood.

ADHD in our adult patients was associated with intellectual disability and social maladjustment. In particular, adult patients who obtained scores indicating ADHD on questionnaires required more social support and clinical care from multiple services than others, suggesting the greater clinical severity related to this disorder. Our investigation also revealed the poor adherence to treatment, with the frequent drop-out of patients with ADHD, which likely conditioned the small size of our final sample, suggesting, at the same time, the difficulty of healthcare services in maintaining continuity in the minor-to-adult age transition.

An enlargement of the sample could have provided more detailed data regarding the chronicity of ADHD or its clinical course in adulthood; however, the very small size of our final sample can represent the difficulty in collecting information and clinical evidence in a real-world health setting regarding the ADHD follow-up. Moreover, further research is needed to highlight the role of drug and/or rehabilitative treatments in preventing the persistence of this disorder and the onset of other clinical problems in adulthood.

Based on our results, we can conclude, in line with the most recent literature, that ADHD should not be considered a disorder confined to childhood/adolescence but rather a potentially persisting condition that associated with other psychiatric disorders and can greatly reduce functioning skills in adulthood. Our study suggests the need for specific training in the adult diagnosis and treatment of this disorder and the implementation of transition protocols from child to adult health services in order to ensure the continuity of care.

## Figures and Tables

**Figure 1 medicina-59-00338-f001:**
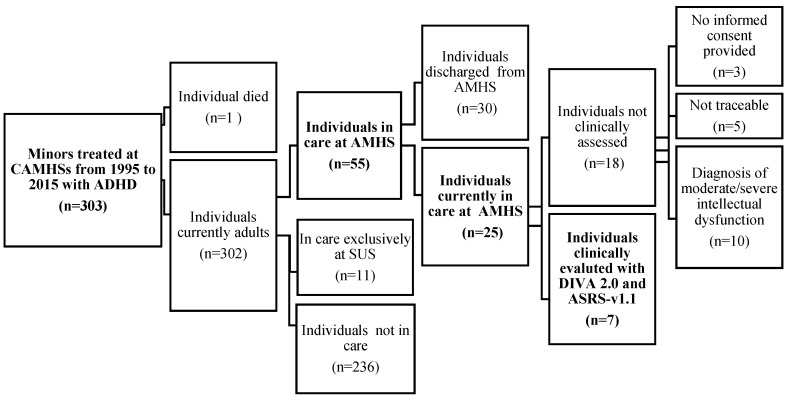
Adult follow-up of ADHD individuals treated at CAMHS in Modena from 1995 to 2015.

**Table 1 medicina-59-00338-t001:** Follow-up of minors in care at CAMHS from 1995 to 2015 for ADHD (n = 303).

Motivations for Conclusion of Treatment at CAMHS	Subjects, n (%)
CAMHS ordinary discharge	187 (78.9%)
CAMHS self-discharge	34 (14.3%)
Referral to AMHS	55 (18%)
Referral to another service (Social Service or Disability Service)	12 (5.1%)
Transfer to another town/country	3 (1.2%)
Deceased	1 (0.4%)

**Table 2 medicina-59-00338-t002:** Variables of subjects (n = 7) divided by DIVA 2.0 score.

Variables	Patients with a Positive Score (≥6) at One or Two Dimensions of DIVA 2.0 (n = 4) (57%)	Patients with a Negative Score (<6) at Both Dimensions of DIVA 2.0 (n = 3) (43%)
**Age at treatment start** (years: mean ± SD)	5.25 ± 2.98	6 ± 0 **
**Referral to CAMHS from**		
School	2 (50%)	2 (75%)
Minors Social Services	1 (25%)	0
Doctor of General Medicine/other specialists	1 (25%)	1 (25%)
**Length of treatment at CAMHS** (years: mean ± SD)	12.5 ± 3.11	10 ± 2.83
**Treatment**		
Rehabilitation	0	1 (25%)
Multimodal (Pharmacological and Rehabilitation)	4 (100%)	2 (75%)
**In care at several services**	4	2 *
**Sex**		
Male	2 (50%)	1 (25%)
Female	2 (50%)	2 (75%)
**Age** (years: mean ± SD)	21.25 ± 1.71	25 ± 1.73 ^§^
**Nationality**		
Italian	3 (75%)	3 (100%)
Not Italian	1 ((25%)	0
**School**		
Secondary school	3 (75%)	0
High school	1 (25%)	3 (100%)
**Employment**		
Employed	0	1 (25%)
Protected job placement	2 (50%)	2 (75%)
Civil invalidity	1 (25%)	0
Unemployed	1 (25%)	0
**Housing**		
Parental Family	3 (75%)	2 (75%)
Alone	1 (25%)	0
Unknown	0	1 (25%)
**Main Psychiatric Diagnosis**		
Intellectual Disability	0	2 (75%)
Personality Disorders	0	1 (25%)
Conduct Disorder	2 (50%)	0
Bipolar Disorders	1 (25%)	0
None	1 (25%)	0
**Period in therapeutic communities**	1 (25%)	0
**In care at Social Service**	3 (75%)	2 (75%)
**Legal issues**		
Yes	1 (25%)	1 (25%)
Legal administrator	0	0

***** Pearson chi^2^ = 3.73; *p* = 0.05; ^§^ t = 2.85; *p* = 0.03 (*t*-test); ** Data of only two individuals were collected.

## Data Availability

Not applicable.

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
