# Peer review of "ADHD Follow-Up in Adulthood among Subjects Treated for the Disorder in a Child and Adolescent Mental Health Service from 1995 to 2015"

_medicina, 2023, doi:10.3390/medicina59020338_

Round 1
Reviewer 1 Report
In the manuscript of “ADHD follow-up in adulthood among subjects treated for this disorder in a Child and Adolescent Mental Health Service from 1995 to 2015,” the authors used a 2-step approach to study the transition and progress of those who were diagnosed and treated for ADHD when they were children/adolescents and when they became adults. This is an important topic and still represents an unmet need.
This was an ambition study. However, because the sample size was too small, it will be difficult to make any inference. For this reason, the current focus will add little information the field.
To make this study a little bit meaning, the authors may consider changing the focus to the unmet need and “lesson learned”.
The authors need some kind of hypothesis of why only a very small number of adult patients received their treatment at their center and try to figure out potential factors associated with this low rate of the continuation of care. In doing so, the authors may be able to propose strategies at different levels to improve the care of patients with childhood ADHD after they become adults.
In the Abstract, the authors need add limitations to the Conclusion.
Introduction is too long. Paragraph 3 and 4 which are related to this study should be deleted.
In Data collection, “Variables collected from AMHS information system: …..” It is unclear how this was carried out. Did some authors or all authors collect those data or other providers collect the data? Where those data collected at the initial evaluation or anytime at the AMHS? Seemingly, there was a standard research protocol in place at collection, but it is unclear how it was carried out? What diagnostic instrument(s) was used? For ADHD evaluation, was the evaluation done only once or multiple times?
In Results, Figure 1 and Table 1 are useful. The rest of the Tables and Figure 2 don’t provide much information and can be deleted. The findings can be described briefly in the text. The authors may consider only presenting the results to help them to support their hypothesis, but not all the results.
One of very striking finding in the sample was a high rate of intellectual disability. Did the authors have the IQ scores of patients diagnose with intellectual disability? It is well known what most patients with ADHD do not have low IQ or intellectual disability. Did the patients in study represent patients with ADHD in Italy? If so, the authors need describe the rate of intellectual disability. If not, the authors need describe how the results in the study can be generated to our patients in Italy or other parts of the world.
The Discuss was also too long. The authors should keep in mind that the validity of the results from this study is very limited because of the very small sample size, and it will be difficult to make any inference. The Discussion should focus on this too.
Author Response
Response to Reviewer#1
R: Thank you for your positive comments and for your valuable suggestions which helped us improve the manuscript.
In the manuscript of “ADHD follow-up in adulthood among subjects treated for this disorder in a Child and Adolescent Mental Health Service from 1995 to 2015,” the authors used a 2-step approach to study the transition and progress of those who were diagnosed and treated for ADHD when they were children/adolescents and when they became adults. This is an important topic and still represents an unmet need.
This was an ambition study. However, because the sample size was too small, it will be difficult to make any inference. For this reason, the current focus will add little information the field.
To make this study a little bit meaning, the authors may consider changing the focus to the unmet need and “lesson learned”.
The authors need some kind of hypothesis of why only a very small number of adult patients received their treatment at their center and try to figure out potential factors associated with this low rate of the continuation of care. In doing so, the authors may be able to propose strategies at different levels to improve the care of patients with childhood ADHD after they become adults.
R: 1) We thank the Reviewer for the valuable suggestions. We reported hypothesis regarding the very small number of adults in our final sample, reporting at the end of Discussion the following paragraphs:
“The very small size of our final sample can represent the difficult transition between child and adult care for complex disorders previously considered uniquely confined to minors, such as ADHD, which survive into adulthood. In particular, for a person with ADHD it can be really difficult to maintain continuity of care due to hyperactivity/impulsivity and inattention that can predispose both to therapy dropout and social- relational maladjustment. The adolescents with ADHD “lost in transition” can undergo negative outcomes in adulthood, as highlighted by many authors [83,84].
Transition care is a "multidimensional and multidisciplinary process" [83], which should address many different needs, including social and educational issues [85]. Health services should provide tailored programs to adolescents with ADHD and their family/caregiver in order to address informational, occupational and social needs and not only pharmacological and psychological treatments [86].”
In the Abstract, the authors need add limitations to the Conclusion.
R: 2) We added limitations to the Conclusion in the Abstract:
“The very small size of our final sample may account for both the high ADHD dropout rate over long follow-up period and the difficult transition from child to adult health care in ADHD treatment.”
Introduction is too long. Paragraph 3 and 4 which are related to this study should be deleted.
R: 3) Thank you for your meaningful suggestion. Paragraphs 3 and 4 were deleted and the Introduction was reduced and greatly modified.
In Data collection, “Variables collected from AMHS information system: …..” It is unclear how this was carried out. Did some authors or all authors collect those data or other providers collect the data? Where those data collected at the initial evaluation or anytime at the AMHS? Seemingly, there was a standard research protocol in place at collection, but it is unclear how it was carried out? What diagnostic instrument(s) was used? For ADHD evaluation, was the evaluation done only once or multiple times?
R: 4) We have tried to clarify this point modifying the paragraph concerning our data collection in the Methods:
“The variables were selected to describe the demographic and clinical characteristics of our sample both in childhood and in adulthood among those available in the information system of outpatient services. Data collected from the CAMHS information system were the following: single parenting, adoption, age at the time of treatment start in CAMHS, referral to CAMHS, possible coexistence of intellectual disability or other comorbidities, types of treatments (pharmacological and not).
We retrospectively collected the following variables from AMHS information system: psychiatric diagnosis and substance use, period of treatment, schooling, employment, housing, hospitalizations, treatments, social service support, legal problems, etc. More information were asked to the referral psychiatrist of the adult individuals treated at AMHS.”
R: 5) We added the following specification in the Limitations in order to clarify this point:
“Furthermore, the data collection of our sample in childhood treatment was limited by the lack of much clinical information, such as the type of drug and/or rehabilitative treatment undertaken in childhood or the scales used for ADHD diagnosis, due to incomplete clinical profiles available in the databases consulted.”
In Results, Figure 1 and Table 1 are useful. The rest of the Tables and Figure 2 don’t provide much information and can be deleted. The findings can be described briefly in the text. The authors may consider only presenting the results to help them to support their hypothesis, but not all the results.
R: 6) Thank you for your helpful comments. We deleted Tables 2, 3 and 4 and we reported the information in the text as suggested. We maintained but greatly reduced Table 5 (in the revised version Table 2). We deleted Figure 2.
One of very striking finding in the sample was a high rate of intellectual disability. Did the authors have the IQ scores of patients diagnose with intellectual disability? It is well known what most patients with ADHD do not have low IQ or intellectual disability. Did the patients in study represent patients with ADHD in Italy? If so, the authors need describe the rate of intellectual disability. If not, the authors need describe how the results in the study can be generated to our patients in Italy or other parts of the world.
R: 7) Thank you for your interesting comment. We have reported our interpretation of this result in the Discussion:
“The high frequency of individuals with intellectual disability in our final sample can suggest that the most fragile and vulnerable subjects, who need the regular support of services for daily activities, maintain the continuity of care more frequently than other individuals with ADHD.”
The Discuss was also too long. The authors should keep in mind that the validity of the results from this study is very limited because of the very small sample size, and it will be difficult to make any inference. The Discussion should focus on this too.
R: 8) Thank you for your important suggestion. We greatly reduced the Discussion and focused it on its very small sample size as reported above.
Reviewer 2 Report
Congratulations on your hard work! I have several minor suggestions.
1. Introduction: The first paragraph can be separated into different sections so readers can follow easily. I am not sure if a paragraph about the pathogenesis is required because it is not discussed in the discussion.
2. Results: Results section is repeating the tables and it might be confusing. I would suggest not sharing the information on the tables and summarizing generally. Also, there is a minor spelling mistake in Figure 1 and the figure can be visually improved.
3. Discussion covers everything. Only the short paragraph about quality of life seems unrelated. I would suggest discarding that.
Author Response
Response to Reviewer 2
Thank you for having appreciated our work.
- Introduction: The first paragraph can be separated into different sections so readers can follow easily. I am not sure if a paragraph about the pathogenesis is required because it is not discussed in the discussion.
Thank you for your meaningful suggestions. The Introduction has been separated into different sections: 1.1. ADHD in adulthood, 1.2. ADHD comorbidities, 1.3. ADHD therapeutic issues. The description of ADHD pathogenesis has been greatly reduced and moved at the beginning of the Introduction.
- Results: Results section is repeating the tables and it might be confusing. I would suggest not sharing the information on the tables and summarizing generally.
Thank you for your valuable indication. The Tables 2,3 and 4 have been deleted and Table 5 (in revision version Table 2) has been reduced. Many information have been reported in the Result section.
Also, there is a minor spelling mistake in Figure 1 and the figure can be visually improved.
Figure 1 has been improved and the spelling mistake has been corrected.
- Discussion covers everything. Only the short paragraph about quality of life seems unrelated. I would suggest discarding that.
Thank you for your comment. The part you indicated has been discarded and the Discussion has been modified.
Reviewer 3 Report
The present study is interesting, as it contains some relevant elements of novelty and investigates a topic of clinical relevance. The manuscript is well written and easy to read. Figures and tables are clear and appropriate.
Overall, the article could represent a worthy contribution to the research field.
Some suggestions to further improve the quality of the manuscript:
Materials and Methods:
Comparisons were performed using t-test but no analysis of the normality of assessed data is reported in the manuscript. An assessment on data distribution (Shapiro-Wilk test) should be performed for all parameters, using non-parametric tests (Mann-Whitney U tests) when a non-normal distribution is observed.
Discussion:
It would be interesting to compare the results of the present study with those of another recent study conducted in a very similar setting (see Valsecchi P. et al., Adult ADHD: Prevalence and clinical correlates in a sample of Italian psychiatric outpatients. Journal of Attention Disorders (2021). 25(4), 530-539.)
The careful diagnostic process for ADHD in adults, including both a screening instrument (ASRS v1.1) and a dedicated diagnostic interview (DIVA) should be mentioned as a relevant point of strength of the present work.
Author Response
Response to Reviewer3
Thank you for your positive comments and your valuable suggestions.
Materials and Methods:
Comparisons were performed using t-test but no analysis of the normality of assessed data is reported in the manuscript. An assessment on data distribution (Shapiro-Wilk test) should be performed for all parameters, using non-parametric tests (Mann-Whitney U tests) when a non-normal distribution is observed.
1) We applied the Shapiro-Wilk test to the data of age and DIVA scores and found a normal distribution. Therefore , we maintain the application of t-test.
Discussion:
It would be interesting to compare the results of the present study with those of another recent study conducted in a very similar setting (see Valsecchi P. et al., Adult ADHD: Prevalence and clinical correlates in a sample of Italian psychiatric outpatients. Journal of Attention Disorders (2021). 25(4), 530-539.)
2) We cited the article you suggested in our data comparison in the Discussion.
The careful diagnostic process for ADHD in adults, including both a screening instrument (ASRS v1.1) and a dedicated diagnostic interview (DIVA) should be mentioned as a relevant point of strength of the present work.
3) In accordance with your suggestion, we mentioned in the limitations and advantage section your comment: “ The careful diagnostic process for ADHD in adults, including both a screening instrument (ASRS v1.1) and a dedicated diagnostic interview (DIVA-2) can be mentioned as a relevant point of strength of the present work.”
Round 2
Reviewer 1 Report
none.